# The Campus as a Smart City: University of Málaga Environmental, Learning, and Research Approaches

**DOI:** 10.3390/s19061349

**Published:** 2019-03-18

**Authors:** Sergio Fortes, José Antonio Santoyo-Ramón, David Palacios, Eduardo Baena, Rocío Mora-García, Miguel Medina, Patricia Mora, Raquel Barco

**Affiliations:** 1Departamento de Ingeniería de Comunicaciones, Andalucía Tech, Campus de Teatinos s/n, Universidad de Málaga, 29071 Málaga, Spain; dpc@ic.uma.es (D.P.); ebm@ic.uma.es (E.B.); rbm@ic.uma.es (R.B.); 2Andalucía Tech, Vicerrectorado de Smart-Campus, Campus Universitario de El Ejido, Universidad de Málaga, 29013 Málaga, Spain; jasantoyo@uma.es (J.A.S.-R.); rmora@uma.es (R.M.-G.); mrmedina@uma.es (M.M.); directora.smart@uma.es (P.M.)

**Keywords:** smart-campus, smart-city, architecture, IoT, remote sensing, education, research initiatives

## Abstract

For the past few years, the concept of the Internet of Things (IoT) has been a recurrent view of the technological environment where nearly every object is expected to be connected to the network. This infrastructure will progressively allow one to monitor and efficiently manage the environment. Until recent years, the IoT applications have been constrained by the limited computational capacity and especially by efficient communications, but the emergence of new communication technologies allows us to overcome most of these issues. This situation paves the way for the fulfillment of the Smart-City concept, where the cities become a fully efficient, monitored, and managed environment able to sustain the increasing needs of its citizens and achieve environmental goals and challenges. However, many Smart-City approaches still require testing and study for their full development and adoption. To facilitate this, the university of Málaga made the commitment to investigate and innovate the concept of Smart-Campus. The goal is to transform university campuses into “small” smart cities able to support efficient management of their area as well as innovative educational and research activities, which would be key factors to the proper development of the smart-cities of the future. This paper presents the University of Málaga long-term commitment to the development of its Smart-Campus in the fields of its infrastructure, management, research support, and learning activities. In this way, the adopted IoT and telecommunication architecture is presented, detailing the schemes and initiatives defined for its use in learning activities. This approach is then assessed, establishing the principles for its general application.

## 1. Introduction

The concept of “Smart-City” appeared as the application of an automatic collection of environmental data and its processing to achieve efficient management of the urban areas, as well as their resources and assets [1]. This approach is supported by the massive application of information and communication technologies (ICTs) and the Internet of Things (IoT) paradigm [2], where a vast number of distributed devices are connected in order to transfer their gathered data. The Smart-City reliance on connectivity points out the importance of telecommunication technologies, as they are just as relevant as the involved sensing and processing activities.

Multiple cities around the world have embraced the Smart-City concept to improve the quality of life and well-being of its citizens, as well as to increase their energy-efficiency, better their governance, and reduce their emissions [3,4,5]. University campuses represent an invaluable opportunity to potentiate this approach, as they concentrate a large community of students, professors, and employees, forming a “population” that is willing to adopt and promote innovations as well as get involved as developers and testers. By applying the Smart-City approach to a university area, the resulting “Smart-Campus” can grow with support by its community and it can also provide huge advantages. Campuses are their own “small cities”, where achievable improvements to its management, governance, sustainability, and learning activities are enormous in terms of energy and water efficiency, emissions, mobility, health and well-being, nature, and teaching. 

In terms of applying the Smart-Campus concept to learning activities, different studies focus on the implementation of high-level smart capabilities that could improve education. For example, the work in [6] presents a brief description of the design of an IoT based Smart-Campus scheme focused on smart parking, smart room, and smart education. An integrated platform for this service is summarily presented, where Wi-Fi is used to connect the different sensors and cameras associated with the platform. However, the envisaged learning-related applications aim mainly at remote learning (e.g., by video streaming). Majeed et al. [7] identify a set of opportunities for incorporating the smart concept to campuses, particularly regarding parking, security, classroom support, and education. In the latter study, the main points identified are also associated with the classroom itself or the online access to learning material.

Other works aim for the smart analysis of and/or support of teaching activities, such as following game-based approaches [8], supporting multimedia conferences [9], students’ health [10], and smartphone applications [11]. More aligned with classical Smart-City applications, different smart approaches (applying IoT and data processing) have been defined for different objectives. In this field, Alvarez-Campana et al. [12] present a university campus IoT platform for environmental and people flows monitoring. In terms of smart mobility, Toutouh et al. [13] develops a mobility prediction mechanism, implementing it on the University of Málaga campus. The work in [14] focuses instead on IoT supported disaster management for smart campuses.

In this context, the University of Málaga (UMA) follows a wide-scope approach to the Smart-City paradigm performed by the development of the UMA Smart-Campus initiative (SmartUMA) [15]. Among its strategic objectives are to provide full support to the management of the environment, to the learning activities of the students, and to the research and innovation of their academic personnel. Firstly, the present paper focuses on the description, as a relevant Smart-Campus example case, of the characteristics, elements, solutions, and key features of SmartUMA. Secondly, it assesses the impact and level of involvement of the campus community in the different research and learning activities, providing recommendations for other universities and infrastructure approaches.

In this way, the work is organized as follows. Section 2 presents the general SmartUMA principles and objectives. Section 3 revolves around the different IoT and cellular infrastructures in the UMA, introducing a general ICT architecture for their coordination and also detailing their direct use in the management of the university and applications for learning and research. Section 4 focuses on specific initiatives for research, innovation, and learning activities promoted by the Smart-Campus as a whole. From the previous points, Section 5 presents the main discussion on the lessons learned from these experiences. Finally, Section 6 presents the conclusions of this work as well as an outlook on future activities.

## 2. UMA Smart-Campus

The University of Málaga was founded in 1972, grouping different preexistent centers. It is a generalist university that includes multiple degrees in different areas of knowledge, such as economics, law, health, sciences, education, tourism, communications, humanities and social sciences, architecture, computer science, industrial engineering, and telecommunications. It has more than 35,000 students, 1200 employees, and 2400 academic staff.

These different areas are divided into different faculties and engineering schools. As its original location (a capture shown in Figure 1a) was close to the city center, from the 1990s onwards, all the new buildings have been constructed in the Teatinos area (Figure 1b), located in the outskirts of the city and covering a surface of 2 Mm^2^.

SmartUMA’s main aim is to apply Smart-City concepts to the campus and to make the University of Málaga a global reference in environmental sustainability. In this sense, the effort of transforming any given city into a Smart-City requires a huge investment. Here, the UMA Smart-Campus aims to demonstrate that the city can become progressively “smarter”. In this way, the campus works as a demonstrator for all cities and particularly as an example case for the city of Málaga.

To achieve this goal, three firmly entangled key areas are addressed—Conservation and construction, sustainability, and application of innovative technologies—As shown in Figure 2. All three establish the foundations for a sustainable, smart, comfortable, and social campus.

As is represented in Figure 3, UMA Smart-Campus objectives include implementing the campus as a Smart-City by itself, providing an urban-lab for the associated approaches, and being a reference of sustainability. The achievement of these objectives also generates new lines in teaching, research, and innovation. These, in turn, are expected to attract talented students and researchers, encourage inter-disciplinary research, offer courses and degrees related to smart-cities, foster the innovation and the creation of new services and products, share standards, open data, and guidelines with the world, as well as turn the campus into a Living Lab.

### Methodology

The actions of this initiative follow six main application categories, or pillars (see Figure 4). Emissions, Energy and Water, Nature and Environment, Health and Well-Being, Mobility, ICTs, and the implication of the University community by Research, Teaching, and Innovation. All of these pillars have a close relationship among each other and must be interpreted as part of a global objective.

While the first four pillars are common categories in the Smart-City concept, the latest two (ICTs and Research, Teaching, and Innovation) have special relevance for SmartUMA and therefore represent the main focus of this work. Hence, the goal of the ICTs is to apply technology to address all the other pillars with different lines of action:Deployment of measuring devices (sensors, cameras, etc.) throughout the campus.Development of telecommunication networks and protocols to communicate the different data sources with open platforms in the cloud.Processing and analysis by means of artificial intelligence and big-data techniques (the information provided by the data sources).Deployment of actuators, which take actions driven by the AI algorithms in response to the data sources.

Moreover, the university implication is the last pillar that further distinguishes the Smart-Campus from a general Smart-City. In this field, the objectives are defined in three areas:Research: To engage researchers in UMA Smart-Campus design and development in multiple fields, including agriculture, ecology, energy management, urban farming, telecommunications, and interdisciplinary approaches in combination with ICTs, such as the work in [17].Teaching: To translate research and teaching into practice on UMA’s campuses by facilitating participation in pilot projects and testbeds for practical lessons. To foster a new generation of environmental leaders providing mentoring, networking, and professional development opportunities. To support the creation of new sustainability-related curricula, programs, and cross-disciplinary opportunities.Innovation: To set up an Open Innovation framework, engaging students, teachers, researchers, and companies. To create new business opportunities concerning Smart-Cities.

Following these lines, UMA annually develops a Plan of Objectives and Environmental Goals that develops annual matches for these strategic lines, setting specific goals and actions to materialize on the campus throughout the year.

Moreover, in order for the UMA Smart-Campus to be feasible, it is necessary to simultaneously work on a series of transverse pillars that are essential for the proper implementation of its objectives. These pillars structure a set of different challenges, actions/methods, and indicators, which form the foundation for the methodologies followed by the UMA Smart-Campus, as presented in Table 1.

## 3. UMA IoT and Telecommunication Infrastructure

The application of ICTs is one of the UMA Smart-Campus strategic lines as support to the rest of its activities. This section describes the proposed and implemented technological architecture for Smart-Campus architectures as well as the details of each of the deployed infrastructures and their applications.

### 3.1. Smart-Campus ICT Framework, Architecture, and Methodology

The general scheme of the SmartUMA infrastructure is shown in Figure 5. Here, a layered framework for Smart-Campus ICT elements and their use is proposed and defined to properly categorize and relate the different elements of a Smart-Campus ecosystem and their interactions.

The different framework layers are defined based on their functionality within the framework. That is, each layer groups the elements with similar functions within the whole. Except for the layer providing the general network (in blue), each layer provides information to the layer immediately above the layer from which it interacts. 

At the bottom of the framework shown in Figure 6, the Sensors, Actuators, and Devices (SAD) layer contains the main elements used for the interaction with the environment, the gathering of measurements (e.g., temperature, humidity), and performing actions (e.g., activate a light, open a water valve). Commonly, the sensors/actuators are deployed as part of different computing and communication platforms—the IoT nodes (also known as sensor nodes or motes). These motes typically integrate multiple sensors/actuators, providing unified memory and computing capabilities, energy supply, casing, and communication systems. 

In a Smart-Campus, different IoT infrastructures (a set of motes and their supporting equipment deployed at the same time and to the same objective) and Radio Access Technology (RAT) are typically used following the rise of diverse needs, technological availability, and research and funding opportunities. The methodology associated with the design and inclusion of a new IoT infrastructure into the UMA Smart-Campus ecosystem follows approximately the common steps of any IoT network (as the ones similarly identified in [18]):To start the process, the opportunity to achieve one or various objectives coming from the Plan of Objectives and Environmental Goals is identified based on the capabilities to overcome the challenges of their associated transverse pillars (as presented in Table 1, e.g., availability of standards and funding, the expected community engagement, etc.).The set of Smart-Campus goals to be addressed is then translated to a specific set of technical and operational requisites. These come from both the specific objectives to be covered (e.g., variables to measure) as well as those defined in terms of security, power supply, telecommunications, legal procurement, and range of application (e.g., minimum number of buildings/area to be covered).From the previous stage, a system model for the planned new IoT infrastructure (to be fitted in the pre-existent UMA Smart-Campus framework) is defined, including its specific hardware and software as well as its level of fulfillment of the requirements established in the previous stages. Depending on the availability of resources and the specificities of the addressed goals, this design might be generated internally or as part of a competitive invitation to tender from external companies.One or several prototypes of the expected nodes and elements to be included in the UMA Smart-Campus framework are tested in order to guarantee its proper behavior, with special emphasis on its connectivity and interoperability with the pre-existent networks and systems. Multiple iterations and limited deployment might be needed for properly adjust the prototypes to their expected integration and performance.Step by step and wide testing are performed to fully evaluate and guarantee the quality of the deployment, which is typically performed in sequential stages of increasing levels of complexity and integration between the different elements and with the pre-existent framework.Final testing, evaluation, and compliance tests are performed for the complete system, classically followed by training sessions for UMA concerned staff and their parallel evaluation of the fully integrated behavior of the deployed infrastructure.

As represented in the figure and further detailed in following subsections, at the current UMA status, three main IoT infrastructures can be identified (marked with vertically dashed rectangles)—a set of smart meters together with an irrigation control system (to be described in Section 3.2) and the UMAIoT/RAT testbed (Section 3.4).

The Operations, Administration, and Management (OAM) layer is where the measurements coming from the different sensors and devices are stored and from where the actions can be launched to the actuators. The different infrastructures commonly have their own dedicated OAM platform. An additional wide-scope OAMSmartCampus platform gathers the data across all the different infrastructures to provide a centralized point for all further activities. 

The connection between the IoT nodes and the OAM is the main task of the elements of the RAT layer [18]. Often, the RATs are specifically deployed for each infrastructure by means of different gateways (GWs). These GWs implement one of the multiple wireless technologies commercially available, e.g., Zigbee. However, depending on the availability of the communications in each area and the capabilities of the motes, the connectivity might be provided directly by the general network (Net.) infrastructure of the campus, or this might serve as backhaul. This general network is typically based on Ethernet and Wi-Fi connections distributed through the campus for their use by students and staff. Alternatively, the link can be provided by a telecommunication operator based on cellular communications or centralized IoT RATs (e.g., Sigfox).

Parallel to these IoT systems is the UMA cellular telecommunication infrastructure dedicated to research, particularly the UMA Heterogeneous Network (UMAHetNet). As is further detailed in Section 3.3, the UMAHetNet consists of a complete Wi-Fi and Long-Term Evolution (LTE) cellular core and picocells. Its main task is to provide a testbed for cellular telecommunication research and learning, giving service to different user equipment (UE) such as smartphones and special testing devices. Moreover, it is also used to provide radio access connectivity to the motes. 

The different elements, their number (Nr.), and the telecommunication access technologies of each infrastructure are summarized in Table 2.

In the following subsections, their various details, characteristics, and applications are described. All heterogeneous systems further detailed in the next subsections are the bases to supporting the activities of the Analysis, Research, and Learning (ARL) layer. Here, the different applications of the Smart-Campus are performed and integrated by students, staff, professors, and external experimenters in the OAMSmartCampus platform in which they are run. Some main initiatives involving the ARL layer are described in Section 4.

### 3.2. Smart Meters and Irrigation Control System

UMA Smart-Campus deployments initiated with a limited set of smart meters, smart sensing, and actuation boxes measuring multiple magnitudes related to the management and maintenance of the Smart-Campus’ electricity, water consumption, well water extraction, fire sensors, and irrigation. 

These devices were equipped with a heterogeneous set of radio access and wired technologies (Wi-Fi, Zigbee, and Ethernet). They were distributed throughout all the campus, each with their own management platform and, in the case of Zigbee, dedicated radio GWs. The data were accessible across a deployment-specific OAM platform, where the information was available in real-time and stored for further processing and analysis.

In addition to these meters, an irrigation control system was implemented. This consisted of multiple stations deployed widely through the campus and equipped with weather and soil-related sensors and caudal control actuators. Data from such sensors allowed optimized irrigation based on measured humidity, temperature, or other physicochemical parameters of the ground. Additionally, it provided an indication of when it was suitable to irrigate the remaining areas not equipped with IoT nodes (based on the common parameters of weather and type of soil), how much water was required, or if plants needed additional care. Depending on their location, the irrigation stations were connected to the general network by different RATs, such as Wi-Fi, cellular (LTE), and/or a low-power wide-area wireless (LPWA) proprietary technology. These stations—also managed by their own dedicated platform—helped to improve the vegetation care and their sustainability, and supported turning most green spaces of the campus into a Living-Lab where students and researchers could investigate vegetation–environment relationships.

### 3.3. Cellular Infrastructure—UMAHetNet

The UMAHetNet is part of the infrastructure of the Communications Engineering department (“Ingeniería de Comunicaciones”, IC). This unique testbed is devoted to the teaching and research on cellular network management, especially in the field of self-organizing networks (SON) [19]. It also supports IoT deployments. 

The IC department has a great deal of expertise in the development of machine learning algorithms for SON. Such a paradigm focuses on the automation of procedures for the management of cellular networks, minimizing both the need for human intervention and the overall operation and capital expenditures while increasing the service performance. Some examples of the procedures being automated are network configuration [20], optimization [21,22], and troubleshooting [23,24,25]. This line of work has fructified on a long relationship with the main stakeholders in the sector and the participation in key research projects. 

The studies performed, however, have been classically limited to simulations, provided datasets, and controlled trials, particularly due to the reluctance of commercial cellular operators to modify their network configurations and risk degrading network performance. To avoid these limitations, the UMAHetNet represents a complete cellular operator’s indoor infrastructure in service of research and teaching activities and compliant with the 3rd Generation Partnership Project (3GPP) Release 9. In this way, it is composed of a fully functional LTE evolved packet core (EPC) containing all the functionalities and network elements for packet routing, mobility handling, and quality of service (QoS) managing (see Figure 7a). Its radio access network (RAN) is comprised of 12 picocells (low power base stations, as presented in Figure 7a), which have been deployed throughout different floors and buildings in the “Escuela Técnica Superior de Ingeniería de Telecomunicación” (ETSIT) from UMA, as shown below. In this figure, the current position of the picocells is presented as a red circle. Blue triangles represent ready-to-plug points—that is, possible locations of where to move a picocell to have a new network layout. This picocell mobility feature allows great flexibility, enabling high interference scenarios as well as high mobility along corridors and between floors. 

Besides their LTE interface, as is also indicated in Table 1, these picocells also provide Wi-Fi (which makes the network heterogeneous), allowing them to act as access points for the Wi-Fi standards IEEE 802.11b/g/n. Together with these, UMAHetNet has an operations support system (OSS) in charge of steadily monitoring and storing network elements performance as well as delivering configuration commands to these. Those activities can be performed based on a management (MGMT) platform through its graphical user interface (GUI) as well as through a command line and direct access to the system databases. Such double management interface has allowed development of a closed loop for network optimization and troubleshooting and will eventually allow for integrating the algorithms for network automatic management developed in the IC department.

Regarding the UEs, UMAHetNet includes 12 commercial smartphones (Huawei P8) and two phone-based drive test tools to measure both RAN performance metrics and service-specific user-centric quality indicators. Besides these, UMAHetNet is also equipped with two testing UEs originated in the framework of the H2020 Research Project Measuring Mobile Broadband Networks in Europe (MONROE) [26]. These devices are special Linux-based “drive test” terminals, which can be remotely programmed (based on Docker technology) to run a variety of services and gather their associated quality metrics. 

To additionally open this testbed to the research community, a representational state transfer (REST) API has been developed to make the functionalities of the UMAHetNet’s OSS accessible from outside UMA. This allows the whole infrastructure to be used as testbed-as-a-service, which has attracted attention in the form of international research projects, like the H2020 European projects MONROE [26] and ONE5G [27], and one funded in the framework of the Huawei Innovation Research Program (HIRP) [28], the HIRPO2017010216 project. In all these research activities, UMAHetNet plays a major role as a testbed platform for the new development in SON and 5G. Moreover, the integration of UMAHetNet into the wider Smart-Campus ICT framework paves the way to combining information coming from the environment into cellular context-aware SON approaches as the ones considered in [21,22,23,24,25].

#### Supported Learning

One key element in the activity of the University of Málaga is the teaching of cellular telecommunications. The presence of main companies of this sector in the Málaga area makes associated learning activities a key task of the Telecommunication School.

The teaching of cellular communications and especially cellular network management have been classically focused on theory classes describing the main elements, protocols, and technologies associated with the technology and its deployment. Although this provides a firm basis for their application in real work, practice lessons are also considered very relevant for the proper training of the students. To support these, different subjects of the telecommunications master studies have recently incorporated the use of UMAHetNet into their planning. In this way, the possibilities of experiencing the management of a real cellular network are available to the students, including the access to the diverse sources of information collected by the network (events, alarms, counters, traces [29]). They can also present the available configuration actions. Moreover, the network can be used to support drive test measurements [30] without data limits or restrictions.

However, providing access for the students includes some challenges. Firstly, the number of computers with access to the Virtual Private Network (VPN) that all UMAHetNet elements are connected to is very limited. Secondly, any possible interference with the research work or damage to the infrastructure has to be avoided. To do so, a multi-layer security scheme is put in place, establishing access restrictions at device, location, user, application, and action levels. Also, accountability of the students’ executed actions is supported. 

This security scheme is presented in Figure 8, which also shows its correspondence to the layers of the proposed Smart-Campus framework in Figure 5. Here, the distinct phases of access control that the user must cross to make use of the network management system can be observed: The lessons are performed in a specific room with desktop computers connected to the department network. The computers, equipped with a Windows-based operative system (O.S.), are accessible using a student account with limited user rights.Once logged into their local O.S., the students must connect to a remote server. This is equipped with two network cards, one connected to the general UMA network and the other to the UMAHetNet VPN.The access to this server is performed through Remote Desktop. The account provided to the students is application (APP)-restricted: when opening the session, it automatically executes the MGMT platform without allowing the student to open any other application. A different remote O.S. account is also made available to the students but is limited to accessing the platform documentation.Accessing the MGMT platform also implies its own user login. This provides access control, the definition of the different level of permissions (the specific monitoring and configuration actions allowed to the user), as well as the storage of the log of past user actions. In this way, the students are also provided with specific credentials, and its actions over the MGMT platform are fully controlled and traced.Moreover, all the provided credentials are valid only temporarily and only in time periods associated with lessons in order to avoid activity outside those hours.

### 3.4. Testbed on IoT and RATs—UMAIoT/RAT

As a natural extension of the pre-existent Smart-Campus infrastructure and the UMAHetNet, a wide IoT deployment has been defined in order to serve as a testbed for both IoT and associated RATs—The UMAIoT/RAT testbed.

This infrastructure revolves around multiple objectives. First, it widely extends the number and variety of the available sensors and actuators in the campus by including additional IoT nodes. These are expected to provide a general boost to all the activities and lines of the Smart-Campus. Secondly, it implements a testbed for telecommunications research in the field of radio IoT-specific technologies—LoRa [31], Sigfox [32] and Zigbee [33], plus Wi-Fi and LTE—Fitting into the general scheme already put in place for the UMAHetNet and the previous IoT deployments. Thirdly, it provides the means to integrate all pre-existent infrastructures of the campus into a common monitoring platform. 

The architecture of this Smart-Campus IoT testbed was designed based on three main levels corresponding to the SAD, RAT, and OAM layers of the framework proposed in Figure 5 and formed by sensors connected to IoT nodes, gateways that provide connectivity to them, and a third level where a centralized platform gathers and processes all the data, where it can be accessed by the users from the above ARL layer.

The sensor network is based on the “Waspmote” OEM (Original Equipment Manufacturer) technology from Libelium [34]. These motes are compatible with a wide variety of sensors, and they provide a fair amount of flexibility, halfway between what would be a prototyping approach (typically based on Arduino, Raspberry Pi, Edison, etc.) and a solution oriented to production environments. In this way, the infrastructure is provided from the start with a wide set of programming libraries, specific use-case assembled nodes, as well as fully functional development Software Development Kit (SDK). This prevents the need for long integration and development periods and allows its immediate use for research, teaching, and Smart-Campus management. 

The more than 120 IoT nodes included in the infrastructure cover a wide variety of configurations of sensors and actuators for different applications: agriculture (terrain temperature/humidity, pyranometer, trunk diameter), events (presence + light + temperature/humidity/pressure [THP] + ON/OFF actuator), gases (CH_4_, O_3_, CO, SO_2_, NO_2_, air pollutants, THP), air particles, radiation, and parking. 

To support the research on multi-RAT protocols and algorithms [35], which is one of the main objectives of this testbed, the motes are equipped with two radio-shield interfaces with different reconfigurable combinations of Wi-Fi, LoRa, LTE, Sigfox, and Zigbee (see Table 2). Spare radio frequency (RF) shields are also available, allowing for quick reconfiguration of the nodes given the experimenter’s needs. 

Following a capillary approach, these nodes connect to different gateways, although P2P connections between the motes are also possible for some of the RATs (Zigbee, Sigfox). The more common gateways are Meshlium, also a solution from Libelium [36]. These contain different radio interfaces for the communication with the motes, including Wi-Fi (at 2.4 GHz and 5 GHz) and Zigbee. For the backhaul from the Meshliums to the rest of the network, Ethernet or cellular communications (4G/3G/2G) can be used. Apart from this, one LoRaWAN gateway is also deployed in the campus (estimated range > 15 km). The Sigfox coverage is provided by a telecommunication operator, while the LTE is given by the UMAHetNet.

All the data collected by the nodes are stored on a management and open-data platform deployed at UMA server facilities. This platform is based on web tools, and it is designed to have a full set of management, monitoring, and control functions for IoT services. Additionally, it is compliant with the European FIWARE open-source platform, offering a flexible approach for the development of new add-ons and features [37,38]. Also, it enables the possibility of data merging from various sources and types, as it accepts all kinds of connectivity solutions. For all this, this platform was chosen to centralize the monitoring of all IoT deployments of the Smart-Campus, as is described in the following section.

### 3.5. OAM Smart-Campus Platform

With the amount and heterogeneity of the different ICT infrastructures described, a shared system for the gathering of data and distribution to the community [1] is deemed indispensable. In this line, the data management platform deployed under the framework of the previous UMAIoT/RAT testbed is used as the central point to integrate all the information coming from the different sources in the campus. As shown in Figure 5, a set of IoT agents provide connectivity to multiple IoT messaging protocols (e.g., Message Queuing Telemetry Transport-MQTT, Constrained Application Protocol - CoAP [39]), making their details transparent for managing the data.

In this way, all gathered information would be stored in an IoT cloud platform based on the aforementioned FIWARE technology [37,38]. This is an open-source platform fostered by the European Union, which allows the connection with IoT, the storage, publication, and analysis of data on a large scale, as well as the co-creation of apps and development of advanced user’s interfaces by using the common standards and information models.

In this way, the platform provides access to real-time context information, which can be consulted or shared for all of the university community for informative or educative purposes, turning the Smart-Campus into a Living Lab.

For the participants of the Smart-Campus, this platform enables them to have a competitive position in research activities. Moreover, it is expected to be a catalyst for the creation of new applications and services supporting the entrepreneurial ecosystem around it, where it is expected to take advantage of the drive provided by FIWARE and its FIWARE Foundation [38]. The latter, dedicated to promoting, augmenting, protecting, and validating the FIWARE technologies, has more than 150 members, including well-known enterprises such as Telefonica, Atos, Orange, Engineering, and NEC. 

## 4. Analysis, Research, and Learning Proposals and Assessment

Traditionally, university education has been mainly based on theoretical master classes and monodisciplinary projects, which might not have a direct application in the student’s everyday life and career [40]. Conversely, the UMA Smart-Campus project aims to support a different methodology—interdisciplinary and problem-based teaching, which is based on cooperative and transversal projects, joining different areas of knowledge to solve problems related to the UMA Smart-Campus pillars. In this way, students and researchers are encouraged to feel like active parts of the university community. Moreover, the participants can learn about other fields in a cooperative and applied manner. To this aim, we take advantage of the existing infrastructures discussed above, which are used as the basis for current and future projects, as well as the possibility of creating new spaces, urban-labs, and platforms. In this way, the projects created are neither isolated nor without connection between them. Conversely, they all have a common link, which favors the growth of the university as a cohesive entity.

Two innovation frameworks to develop this kind of project are presented in the following sections, “Green Islands and Routes” and “Smart-Campus Innovation Plan”. For both, the involvement of the university community is assessed at multiple levels, providing insight to the general interests and motivation in the Smart-Campus field. 

### 4.1. Green Islands and Routes

A green island is defined as an area situated close to the different buildings of the campus that provides solutions to the limitations of the specific center/faculty to which it belongs. It is characterized by its vegetation, energy efficiency, advanced technology, and comfort. Its purpose is to provide places to stay outdoors to study, meet, rest, etc. The main peculiarity of these places is that they are designed by students supported by the teaching, research, and operational staff of the university. The different areas suitable for adaptation as green islands have been defined across the campus, as shown in Figure 9. 

The methodology to design the green routes and islands is based on two main objectives, the first of which is to satisfy the needs of the users who use the building. To do so, the requests and suggestions of the main expected users of these installations are collected. In addition, some basic requirements are also considered—the space must be sustainable, integrated in the campus, technological, dynamic, and accessible. Secondly, the approach aims to generate a pedagogical, multidisciplinary, and cooperative process where everyone learns from designing and developing the green island. To achieve this, a call is advertised to all the university community to join the project on a voluntary basis. Students, teaching and research staff, as well as the operational staff of the university work together to design this new area based on the knowledge and experiences of each one of them. Furthermore, they have the motivation of knowing that the space they are designing will be built in the university. 

To reach these objectives, the development of the green islands is divided into different phases:Search for requirements and preferences: As a starting point, the users of the buildings where the green islands will be built are surveyed about their preferences and requirements. To perform this step, the survey is disseminated online through the official website of the university, social networks, as well as randomly distributed face-to-face to people who belong to the personnel and users of the buildings involved.Creation of groups: Participation in the further steps of the activity is widely promoted to attract participants. Students from different disciplines (science, geography, architecture, computer science, etc.) are divided into multiple interdisciplinary teams. Supervising professors are assigned to each team in such a way that each of them has an expert in each branch of knowledge who supervises, inspires, and guides them. Furthermore, the professors are encouraged to integrate the project into their courses or as the base for a bachelor/master thesis.Design: Each team is responsible for a different island or route where they compete among each other for the best designs.Evaluation: A tribunal, formed by members of the Vice-Rectorate of the Smart-Campus representatives of the different faculties and participants from previous years, evaluates the different projects. The two best groups win a prize.Implementation: All designed projects are built, favoring the development of the campus and the motivation of the university community since they build their own university based on their needs and knowledge.

The scope of the project includes structural and architectural solutions (design of the spatial planning, climatically adequate spaces and shadow structures, and improvement of accessibility and usability), design of urban amenities based on recycled materials, monitoring and sensing system, choice of vegetation, study of the ground, and environmental parameters. Moreover, the topographic, climatic, and sustainable requirements as well as the end user’s needs are taken into account.

#### Involvement Assessment

The following figures present the participation in this project during the course 2017/2018. Firstly, in Figure 10, the involvement of the different sectors in the surveys about requisites and preferences for the green islands/routes is presented and shows an important involvement by the students but also of all university personnel.

The result of the surveys indicated mostly that both students and workers of the university demanded shaded areas where they could rest in summer and climatically adequate areas to allow them to stay in the open air in winter. Moreover, students requested a leisure area where they could relax, exercise, or practice some sport, such as ping pong or basketball. Another interesting result was the request for a place to teach outdoor classes by teachers.

With regards to participation in the design and support of the projects, the number of involved professors (who worked on the project altruistically) was 38. Figure 11 demonstrates the interdisciplinary nature of this project; there was involvement of eleven fields of knowledge, and geography, architecture, and education sciences were the more numerous areas.

As for the students (see Figure 12), 76 took part in the project, and there were nine different fields of knowledge. The number of participants from telecommunications engineering, geography, and architecture stood out, with 21, 19, and 13, respectively. The results show a proper interdisciplinary reception by the UMA community.

### 4.2. Smart-Campus Innovation Plan

In 2018, the Vice-Rectorate of the Smart-Campus launched its first Smart-Campus Innovation Plan. The main aim of this grant call is to foster research, innovation, and development of interdisciplinary projects based on the Smart-Campus infrastructures and framework. The topics of these projects must be related to the pillars of the UMA Smart-Campus (as presented in Section 2 and Figure 4), and they must include an operative prototype in the campus at the end of the project.

Since the execution of multidisciplinary projects with the involvement of the university community is a priority, a requirement for the projects includes that the participating teams must be formed of at least two subgroups of different areas of knowledge. Also, each subgroup should include at least two students and a principal investigator. External organizations and companies are also encouraged to participate in the teams. The granted funding can be dedicated to both staff and prototypes. The cost of staff could be used for hiring dedicated personnel and for grants for UMA students. 

Figure 13 presents the participation data, which were high with 58 project proposals and 724 people involved. Teaching and research staff were the most numerous participating sectors, 52% of the total, followed by students, 32%. Another remarkable detail was the high participation of women (36%) even though the fields of knowledge with larger participation were sectors where the percentage of men tends to be larger [41,42,43].

The project proposals deal with topics related to one or more of the UMA Smart-Campus pillars. The top lines are “ICT”, “Emissions, Energy, and Water” and “Research, Teaching, and Innovation”, with 37, 36, and 34 projects related to them, respectively (see Figure 14).

The interdisciplinary nature of the projects, which was one of the main goals of this plan, was successfully achieved because more than 17 areas of knowledge participated (Figure 15). Industrial engineering was the sector with the greatest involvement with 219 participants (a 30%), followed by architecture (101 participants, 14%), computer science (98 participants, 13%) and sciences (biology, chemistry, and associated studies) with 78 participants (11%).

In this first edition, 15 projects of the 58 presented are funded. The total budget is €252,000. Table 3 is a summary of the selected projects. They are being developed in the 2018–2019 academic year and will have an operational prototype on campus by the end of it.

These projects display the interdisciplinary nature of the funded projects as well as the wide variety of research and applications that the Smart-Campus can provide.

## 5. Discussion

The deployments and infrastructures of a Smart-Campus might be performed under multiple projects and funding opportunities spanning years, leading to the final coexistence of very heterogeneous ICT elements. This condition presents a challenge in maintaining the proper integration of all the different equipment. To overcome such an issue and to properly allow full exploitation of the gathered data, a proper framework and a centralized IoT management and data system is deemed indispensable.

Furthermore, the management and coordination of the coexisting telecommunication technologies also introduce specific challenges. The compatibility when sharing the radio spectrum must be guaranteed. Moreover, the access to the IoT nodes shall also be independent of the specific radio technologies used with it. Additionally, the management of the different OAM systems implies additional resources need to be dedicated in their proper and coordinated use. 

The planning, deployment, maintenance, and exploitation of a broad and heterogeneous Smart-Campus system is an arduous process that involves multiple departments and institutional groups, such as the different Vice-Rectorates, maintenance staff, schools/faculties directors, teaching personnel, and students. Additionally, it needs the support of equipment providers and deployment companies. The labor of coordinating the different tasks and associated stakeholders requires a tremendous amount of effort, where dedicated and trained personnel is a must. The analysis of the participation in the various activities reflects a significant interest in the promoted ones, especially if resources are provided for research purposes focused on the field of Smart-Campus.

The capabilities and opportunities provided by the Smart-Campus infrastructures must be adequately disseminated to all the stakeholders, both internal (researchers, professors, students, maintenance personnel, employees) and external (research partner institutions, companies, and the public) to take full advantage of its different elements as well as to properly guide its evolutions. 

From a managerial perspective, the monitoring and guidance of the different activities and projects generated from the Smart-Campus (with a very heterogeneous nature and objectives) introduce a new level of complexity and effort that adds to the already challenging technical management. To overcome this, dedicated, technical, and widely knowledgeable personnel is required for the proper coordination and exploitation of the results of the different deployments, activities, and research projects. In this line, some level of competition between the different project groups might be encouraged based on their annual evaluations, access to additional resources, and their visibility in the dissemination activities. All of this combined with the highest possible level of collaboration and transparency between different teams with allow participants to fully take advantage of the common synergies, especially in the use of the environment and in the integration in the general UMA Smart-Campus framework.

## 6. Conclusions

In this work, the principles and driving characteristics of the University of Málaga Smart-Campus approaches in Smart-City learning and research activities are detailed. A general framework of the different layers of a Smart-Campus is presented that describes the main technological infrastructures associated with their implementation. This shows the heterogeneity of this kind of infrastructure and the challenges it presents.

The different ongoing innovation and educational activities associated with the Smart-Campus are detailed, analyzing their impact on the university community. We assess the interest generated by the Smart-Campus formula in people from many different disciplines and how different interdisciplinary collaborations are encouraged by the proposed activities (green islands and routes, Smart-Campus Innovation Plan). However, for the Innovation Plan projects, the involvement of participants from architecture and engineering studies is higher than other studies. Further editions will aim to increase the awareness and interest for other studies that can greatly benefit from the analysis of the gathered data and that seem underrepresented at the moment, such as medicine, economics, and social sciences. 

Finally, different lessons are discussed as technical and managerial guidance for further Smart-Campus deployments and associated innovation and educational activities.

The UMA Smart-Campus initiative will continue growing in the short-term with the expansion of its current infrastructure, the implementation of the green island, innovation plan proposals, and the application of the gathered data in the improvement of the management of the campus. From a technical perspective, future works will address the challenges and lessons learned from the ever-growing addition of new infrastructure to the defined UMA Smart-Campus common framework, which will eventually reach far beyond its current dimensions. From a managerial point of view, further studies will take advantage of the increasing history of activities, projects, and polls to be analyzed, providing a deeper understanding of the impact of the Smart-Campus approach in the community as well as allowing for the analysis of its temporal evolution and the impact generated by the different policies and the specific project being adopted.

In summary, the development of the UMA Smart-Campus with the ultimate aim of sustainability will lead to an improvement in the administration, management, and decision making of the university due to a greater knowledge of all the information that surrounds it. In addition, all these advances increase the quality of life of the university community and allow the development of new areas of interest. It is also giving rise to a methodology of learning, study, and innovative work that increases the skills, commitment, and knowledge of all those who make up the university, turning it into a unique and pioneering place whose methodologies can be examples to be replicated in other universities and organizations.

## Figures and Tables

**Figure 1 sensors-19-01349-f001:**
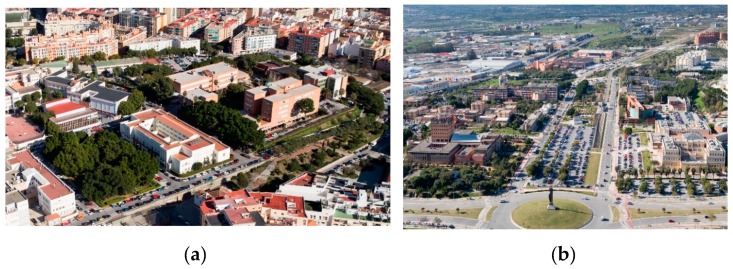
Captures of the section of the campus located at the city center (**a**) and the Teatinos Campus (**b**) [16].

**Figure 2 sensors-19-01349-f002:**
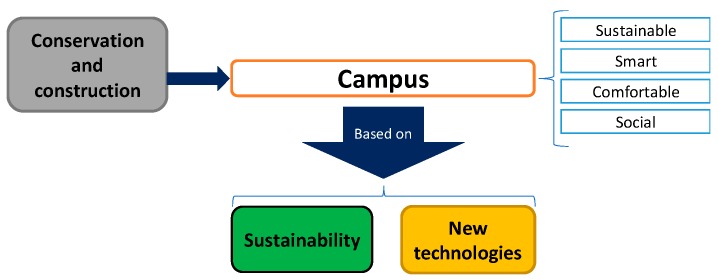
University of Málaga (UMA) Smart-Campus areas.

**Figure 3 sensors-19-01349-f003:**
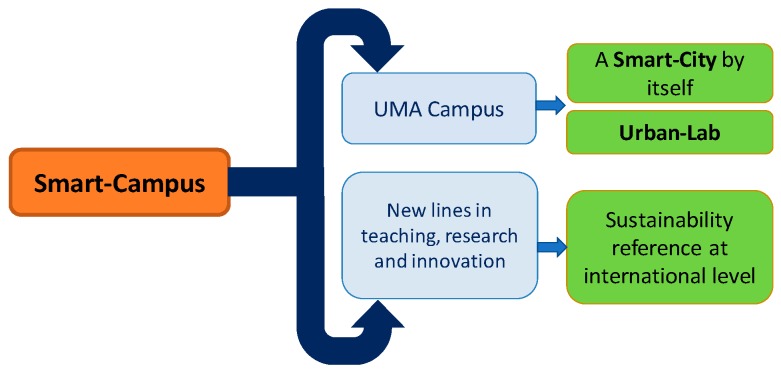
Objectives.

**Figure 4 sensors-19-01349-f004:**
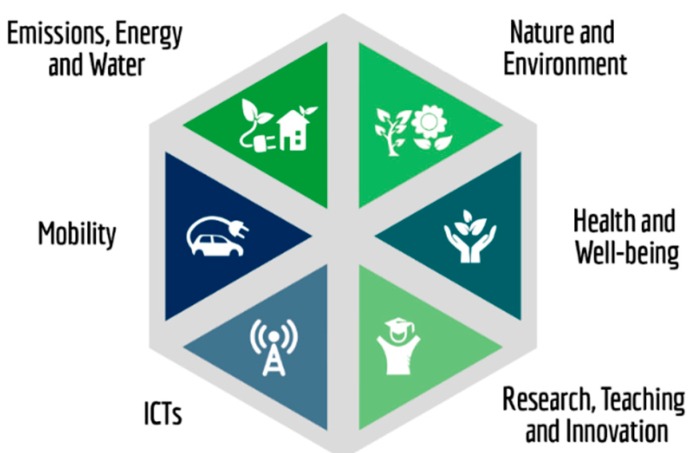
UMA Smart-Campus (SmartUMA) application categories (pillars).

**Figure 5 sensors-19-01349-f005:**
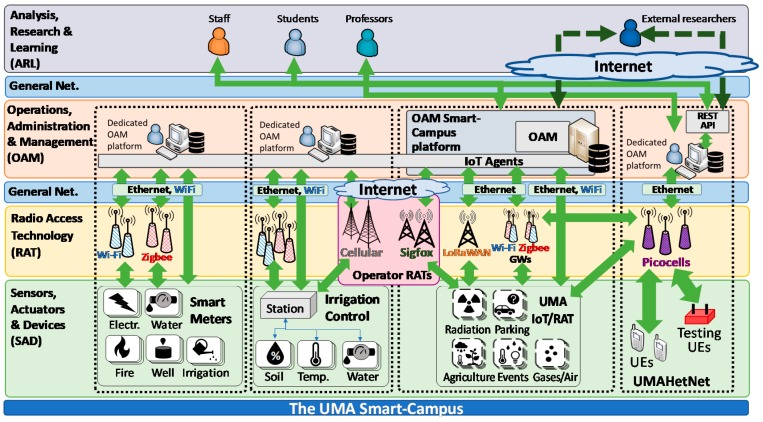
UMA Smart-Campus framework and general scheme of the different information and communication technologies (ICT) infrastructures.

**Figure 6 sensors-19-01349-f006:**
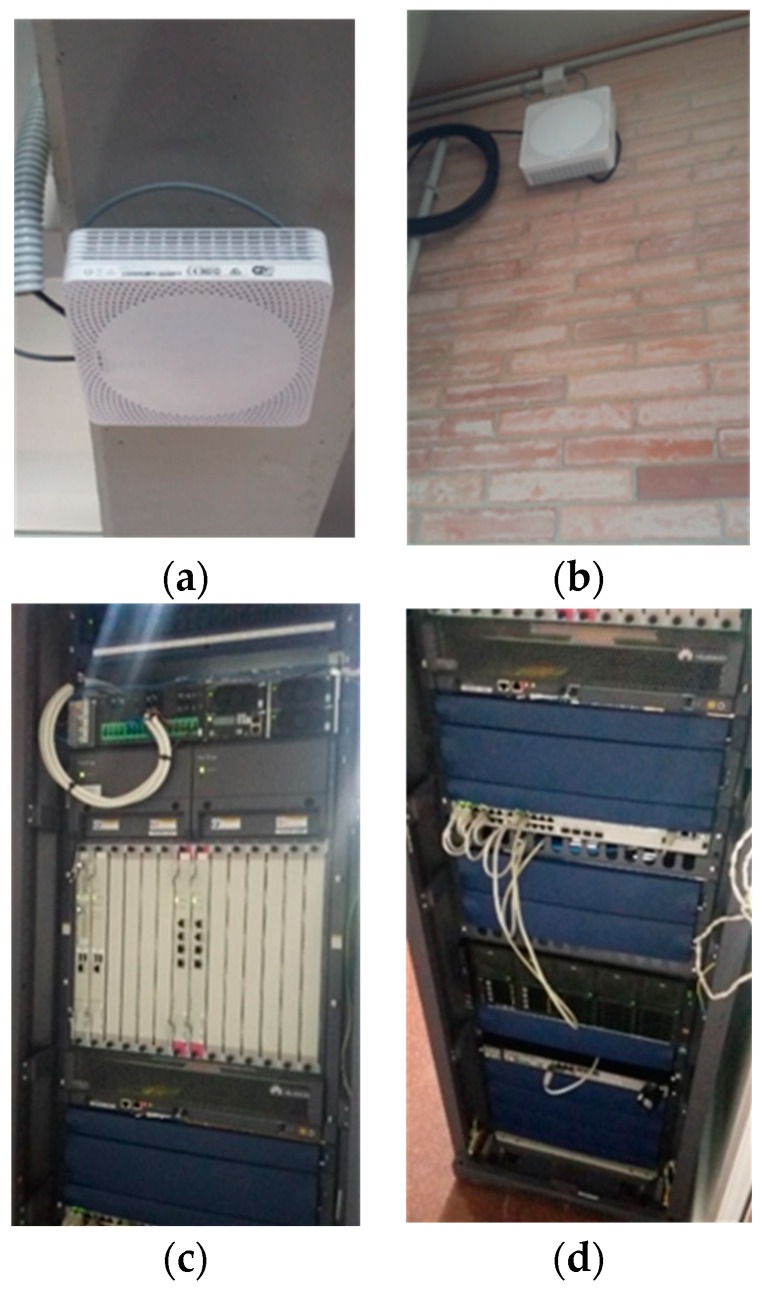
UMAHetNet equipment captures: deployed picocells (**a**,**b**) and UMAHetNet’s EPC (**c**,**d**).

**Figure 7 sensors-19-01349-f007:**
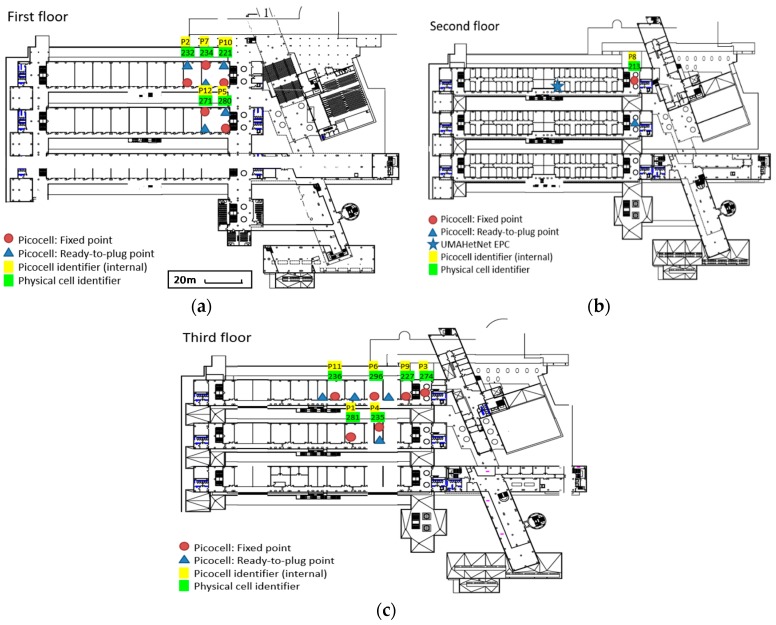
Picocell deployment in “Escuela Técnica Superior de Ingeniería de Telecomunicación” (ETSIT) throughout different floors.

**Figure 8 sensors-19-01349-f008:**
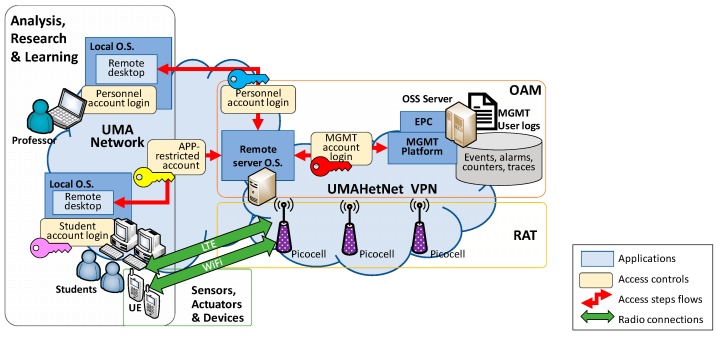
UMAHetNet learning access scheme.

**Figure 9 sensors-19-01349-f009:**
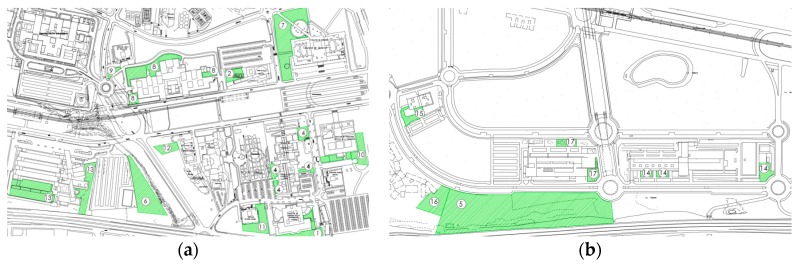
These maps present the campus of University of Malaga, (**a**) is the area denominated Teatinos, (**b**) is the extension of Teatinos (latest zone). The numbered spaces colored in green are the selected places for turning into green islands.

**Figure 10 sensors-19-01349-f010:**
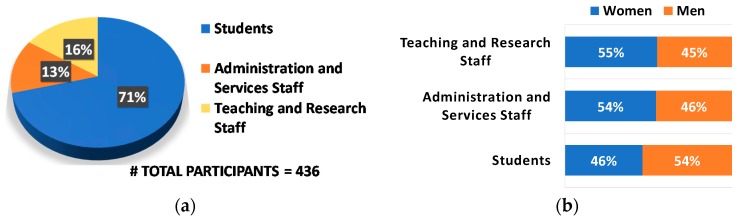
Participation in the surveys on users’ requirements per university community sector (**a**), per gender (**b**).

**Figure 11 sensors-19-01349-f011:**
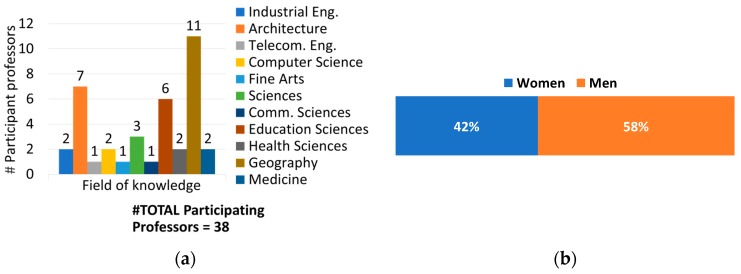
Professors in the project in relation to their field of knowledge (**a**) and gender (**b**).

**Figure 12 sensors-19-01349-f012:**
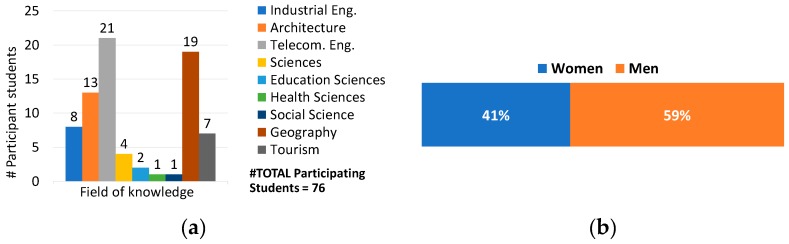
Number of participating students in relation to their area of knowledge (**a**) and gender (**b**).

**Figure 13 sensors-19-01349-f013:**
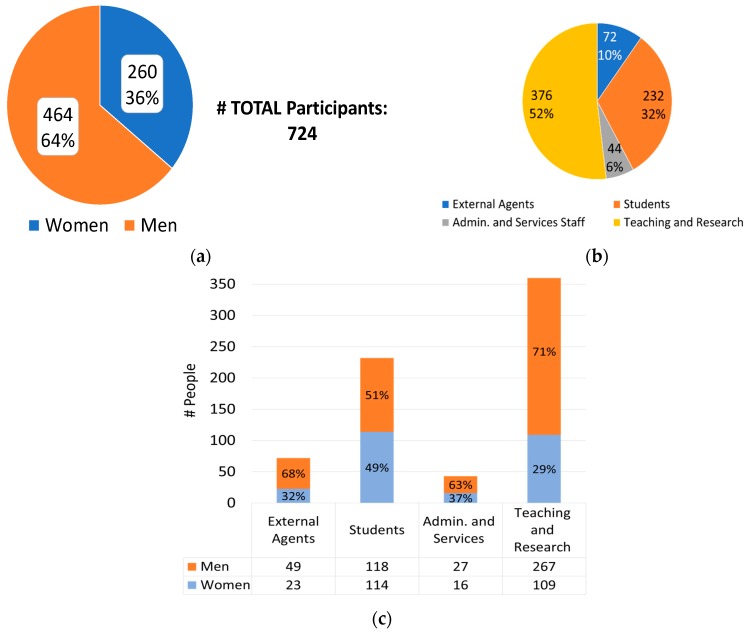
Number of participants in relation to gender (**a**), sector (**b**), and both (**c**).

**Figure 14 sensors-19-01349-f014:**
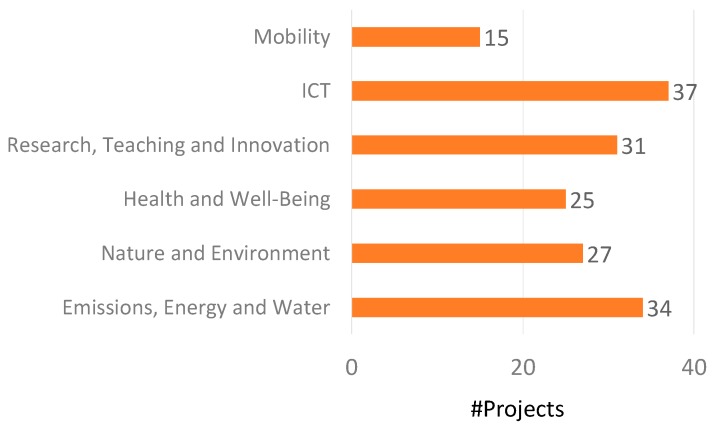
Number of projects in relation to the UMA Smart-Campus pillars.

**Figure 15 sensors-19-01349-f015:**
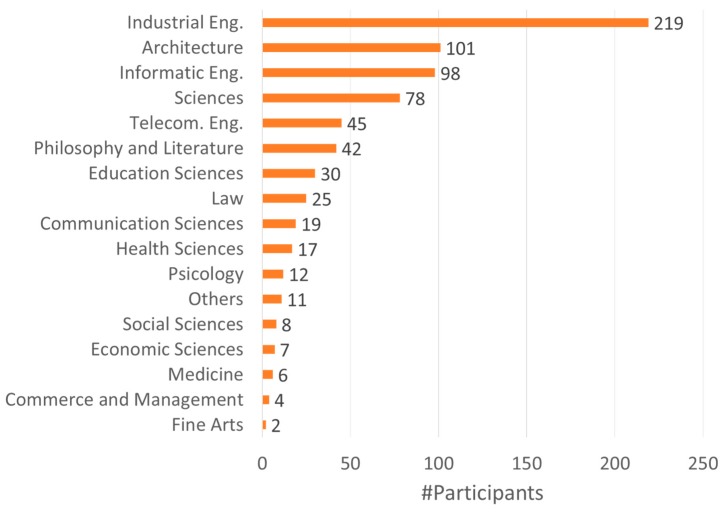
Participants in relation to their field of knowledge.

**Table 1 sensors-19-01349-t001:** Transverse pillars, challenges, actions/methods, and indicators.

Transverse Pillars	Challenges	Example of Actions/Methods	Example of Indicators
Standards	Development and use of standards and guidelines to be applied in Smart-Cities based on the proven success in UMA Smart-Campus	Creation of a team of experts Development of standards and guidelines to follow	Number of standards Success rate
Funding	Creation of a financing plan that includes internal and external funds	Search local, national, and international support to obtain funds that allow development of some of the UMA Smart-Campus actions Involve large companies and small and medium-sized enterprises (SMEs) that might invest in UMA Smart-Campus projects	Budget intended for UMA Smart-Campus projects.
Engagement	Commitment of the university community and agreements with external companies	Creation of a community of partners	Number of external partners Number of members of the university community
Dissemination	Raise awareness on UMA Smart-Campus activities and sustainable commitment to educate, involve, and motivate the community	Creation of teams composed of teachers and students to help develop each pillar of the campus Launch of awareness campaigns	Number of visits to the website and social networks Number of contributions of the members of the community of UMA
Living-Lab	To make any UMA Smart-Campus action an opportunity to investigate and share the knowledge developed with the rest of the world	Developments of prototypes in the campus	Number of created prototypes
Data and Supervision	Obtain the maximum possible campus information to make decisions and publish the progress in relation to the set of objectives	Establish a network of sensors and other suitable devices that allow obtaining real information about the environment and what is happening in the campus Create an integrated auditing system in the UMA Smart-Campus	Number of sensor systems designed and installed Number of goals completed

**Table 2 sensors-19-01349-t002:** Internet of Things (IoT) main elements and communication technologies.

Infrastructure	Elements	Nr.	Access Technology
Smart Meters	Electricity Meter	26	Ethernet, Wi-Fi, ZigBee
	Water Meter	24	Ethernet, Wi-Fi, ZigBee
	Fire Sensor	11	Ethernet, Wi-Fi, ZigBee
	Well Extraction	1	ZigBee
	Irrigation	6	Ethernet, Wi-Fi, ZigBee
Irrigation Control	Exterior Station	23	Wi-Fi, LPWA (proprietary), Cellular (LTE)
	Sensors (temperature, soil …), actuators	-	Wired connection to the Station Programmer
UMAHetNet	Picocells	12	LTE (Rel 9, 5–20 MHz in 2.1 and 2.6 GHz bands), Wi-Fi (2.4 GHz band)
	EPC	1	Ethernet
	Smartphones	12	LTE (Rel 9), Wi-Fi
	Drive test terminals	2	LTE (Rel 9), Wi-Fi
	MONROE nodes	2	LTE (Release 9), Wi-Fi
UMAIoT/RAT	Motes: agriculture, events, gases, air, radiation, and parking	120	LTE + Wi-Fi LTE + ZigBee LTE + LoRaWAN LTE + Sigfox Wi-Fi + LoRaWAN
	Gateways	10	Mote access: Wi-Fi, ZigBee Backhaul: Ethernet, cellular(4G/3G/2G)
	LoRa gateway	1	LoRaWAN

Long-Term Evolution (LTE), UMA Heterogeneous Network (UMAHetNet), Radio Access Technology (RAT), evolved packet core (EPC), low-power wide-area wireless (LPWA), Measuring Mobile Broadband Networks in Europe (MONROE).

**Table 3 sensors-19-01349-t003:** Description of the selected projects in the Smart-Campus Innovation Plan.

Acronym	Description
APICAMPUS	The proposed work consists of developing a pilot project for the installation and monitoring of beehives in the UMA Smart-Campus. A pollen characterization of different “urban” honeys will also be made, and the properties of bee products (honey and propolis) will be studied. It is noteworthy that environmental dissemination will be made to raise awareness in society about the importance of bees and other pollinators in cities and achieve a more sustainable urban environment. It is also intended to monitor the hives, and the data from the sensors would help in knowing more about the behavior of bees inside cities, which is important in the sustainability of urban ecosystems.
Biblio-Smart	Currently, the limited number of study places in the libraries, computer rooms, and other cultural spaces of our campuses, the potential number of users, and the distance between them generates a huge scarcity in study places. This creates the need to offer the certainty to the user that, after making a trip to the study facility, the user will find a free space. By using novel techniques, it is proposed to develop and manage a patentable device consisting of linking Building Information Modeling (BIM) models of cultural spaces with free entry and requiring physical reservation of spaces. All through an IoT-based system connected to an APP, which will allow the identification of the potential user through its mobile.
CAI_UMA	This project intends to perform the following actions: to measure, monitor, and compute the indoor air quality through the development and integration of measurement instrumentation and communication systems, as well as the treatment of the registered data in order to perform diagnosis of the parameters of indoor air quality. The magnitudes to be measured are temperature, relative humidity, and levels of CO, CO_2,_ and radon. This will make UMA the first Spanish university to incorporate radon measurements in an indoor air quality study.
CIES-C	The objective of this project is to define an intelligent control algorithm that integrates the operation of the shadow elements and the building’s air conditioning system, optimizing energy consumption and ensuring thermal and visual comfort (prioritizing the entry of natural light).
CONMET	The objective of the project is to automatically connect the meteorological station installed in the campus with the irrigation control system currently used by UMA. The data from the station would be processed along with the information received from a wireless sensor network. In this way, the process will be optimized, improving efficiency and sustainability in the campus.
DIAS2P + StreetQR	The problems addressed with this project are increasing the safety of pedestrians in pedestrian crossings that do not have traffic lights (when most accidents occur) and capturing and transmitting information of vehicular and pedestrian flows.
E4 (SmartCity Kids + AulaVerde + Smart Parametric Pavilion)	The aim is to create a prototype for the semi-exterior classroom (non-confined space) on the campus. Thermal and aeraulic simulation models will be created, which will help to decide the configuration of the space, the terminal units, and the thermal production system. The prototype project includes the use of parametric design and some variability of adaptability with integration of the systems assets, sensors, and environmental intelligence. An adaptive urban design that evolves with time and new learning environments will be created, and these spaces will also be used for the care and attention of children.
FRATERNI-LAB SMART-UMA	The need that this project identifies is improving the level of happiness. Given the growing configuration of social relationships as cybernetic networks, and in parallel with the efforts of economic and technological development, the World Reports on Happiness of the United Nations shows the need for sustainable development, taking into account social and ecological aspects. It is those two aspects that this project focuses on, offering the creation of a laboratory that, in a very interdisciplinary way and with the help of researchers from social sciences and experimental and technological sciences, will offer guidelines for the improvement of happiness by improving interpersonal relationships and human–nature ones.
GREEN-SENTI	The proposed solution in this project consists of a new web service for the monitoring of the green areas of the campus and its evolution in general through the capture and analysis of satellite images Sentinel-2 of the Copernicus program of the EU. This service will be implemented as a demonstration pilot to support decision-making both for maintenance personnel and for management and planning. The data engine will be carried out, keeping in mind that it is to be scaled to the frame of the city of Malaga as well as to other universities and cities.
MAHDUMA	This project proposes the manufacturing of a micro olive mill. Here, it is intended to take advantage of the olive production in the campus to generate oil of the UMA brand. The prototyping and manufacturing of the micro olive mill and the analysis of oil quality will be the main objectives. The final design might be commercialized for sale to small olive farms and cooperatives.
Secure EV-Urban Lab	The ultimate goal of this project is to provide ICT support and provide the campus with a pilot infrastructure for the management of sustainable mobility, promoting the use of the campus as an “urban-lab” to carry out ambitious projects where they can test new ideas and innovative concepts related to electric vehicle charging infrastructures and energy management systems of Smart-Cities. To do this, within the framework of the Smart-Campus of the University of Malaga, an open laboratory composed of intelligent multimodal charging points and bidirectional for electric vehicles will be defined.
Smart Trees: Reusing UMA Waste	The purpose of this project is the development of a technological tree prototype built by means of a removable and transportable system assembled from reused materials. Smart-Trees should be scalable elements valid for both the interior of university buildings (lobbies, courtyards, and paths) where the tree includes the elements of support and irrigation of natural vegetation, as well as creates spaces for shade, rest, and work exterior. The Smart-Trees will integrate systems for the generation of renewable energies (wind, photovoltaic) and will create points or technological nodes with WI-FI connection, mobile recharge, computers, and electric bikes, which will integrate temperature, humidity, air quality, and noise sensors.
UsMArtDrive	The objective of this project is the development of tools to allow the collection of data and characteristics of the vehicles that access the University Campus of Teatinos every day with a double main objective. First, the characterization of the most frequent driving patterns between different places of residence and the campus, as well as within the campus itself, and second, the analysis of the data obtained for the generation of information that provide feedback to the driver and to the administration in such a way as to optimize the use of vehicles and improve traffic in and around the campus.

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
