# Peer review of "The Campus as a Smart City: University of Málaga Environmental, Learning, and Research Approaches"

_sensors, 2019, doi:10.3390/s19061349_

Round 1
Reviewer 1 Report
This paper focuses an interesting topic . Purpose is clear, literature is reach and updatedy.
Anyway it lack of a paragraph that describes methodology and this should be added.
Results are interesting. Conclusions should be improved and I suggest to add also managerial implications and limits of the research
Author Response
First, the authors are very grateful to the reviewers and editors for the helpful suggestions and valuable comments to improve the revised manuscript. A point-by-point response to the comments and recommendations is attached in pdf. Also, the manuscript has been updated from its previous version.

Reviewer 2 Report
Paper presents developments and use cases of a smart city/iot deployment program at the context of a university.
The contributions of the paper are not core technical (although the tech stack of various tested elements is described) but are revolving around innovation and educational activities - elements that are quite important for the sustainability and maintenance of infrastructure.
it would be great if the authors go deeper in the experiments and present some . of the key results - present some of the findings based on data facts - what data science methods used and so on.
Author Response

(The authors gave the same response as above.)
